# Promoting inter-organisational knowledge sharing: A qualitative evaluation of England's Global Digital Exemplar and Fast Follower Programme

Susan Hinder[1], Kathrin Cresswell[2], Aziz Sheikh[2], Bryony Dean Franklin[3], Marta Krasuska[2], Hung The Nguyen[1], Wendy Lane[4], Hajar Mozaffar[5], Kathy Mason[4], Sally Eason[4], Henry W. W. Potts[6], Robin Williams[1]*

1 Institute for the Study of Science, Technology and Innovation, School of Social and Political Science, The University of Edinburgh, Edinburgh, United Kingdom, 2 Usher Institute of Population Health Sciences and Informatics, The University of Edinburgh, Edinburgh, United Kingdom, 3 School of Pharmacy, University College London, London, United Kingdom, 4 National Health Service Arden and Greater East Midlands Commissioning Support Unit, Warwick, United Kingdom, 5 Business School, The University of Edinburgh, Edinburgh, United Kingdom, 6 Institute of Health Informatics, University College London, London, United Kingdom

* r.williams@ed.ac.uk

## Abstract

### Background

The Global Digital Exemplar (GDE) Programme was designed to promote the digitisation of hospital services in England. Selected provider organisations that were reasonably digitally-mature were funded with the expectation that they would achieve internationally recognised levels of excellence and act as exemplars ('GDE sites') and share their learning with somewhat less digitally-mature Fast Follower (FF) sites.

### Aims

This paper explores how partnerships between GDE and FF sites have promoted knowledge sharing and learning between organisations.

### Methods

We conducted an independent qualitative longitudinal evaluation of the GDE Programme, collecting data across 36 provider organisations (including acute, mental health and speciality), 12 of which we studied as in-depth ethnographic case studies. We used a combination of semi-structured interviews with programme leads, vendors and national policy leads, non-participant observations of meetings and workshops, and analysed national and local documents. This allowed us to explore both how inter-organisational learning and knowledge sharing was planned, and how it played out in practice. Thematic qualitative analysis, combining findings from diverse data sources, was facilitated by NVivo 11 and drew on sociotechnical systems theory.

**Data Availability Statement:** Data cannot be shared publicly because strict confidentiality commitments were made concerning anonymization of research sites and respondents as a condition of being given access. Although transcripts are de-identified, in some instances it would be possible to identify the organisation and the interviewee from the discussions in the interview on the specific IT systems in the organisation. Data are available from upon request to School of Social and Political Science Research Ethics Committee, University of Edinburgh for researchers who meet the criteria for access to confidential data. The contact details are ethics-stis@ed.ac.uk.

**Funding:** This article has drawn on a programme of independent research funded by NHS, NHS England OJEU Contract reference number 2017/S 114-229670 (RW). The views expressed in this publication are those of the authors and not necessarily the NHS or the Department of Health and Care. The funders had no role in study design, data collection and analysis, decision to publish, or preparation of the manuscript.

**Competing interests:** All authors are investigators on the evaluation of the Global Digital Exemplars Programme (http://www.ed.ac.uk/usher) AS was a member of the Working Group that produced "Making IT Work" and was an assessor in selecting GDE sites. BDF supervises a PhD partly financed by Cerner, unrelated to this paper. This does not alter our adherence to PLOS ONE policies on sharing data and materials.

## Results

Formally established GDE and FF partnerships were perceived to enhance learning and accelerate adoption of technologies in most pairings. They were seen to be most successful where they had encouraged, and were supported by, informal knowledge networking, driven by the mutual benefits of information sharing. Informal networking was enhanced where the benefits were maximised (for example where paired sites had implemented the same technological system) and networking costs minimised (for example by geographical proximity, prior links and institutional alignment). Although the intervention anticipated uni-directional learning between exemplar sites and 'followers', in most cases we observed a two-way flow of information, with GDEs also learning from FFs, through informal networking which also extended to other health service providers outside the Programme. The efforts of the GDE Programme to establish a learning ecosystem has enhanced the profile of shared learning within the NHS.

## Conclusions

Inter-organisational partnerships have produced significant gains for both follower (FF) and exemplar (GDE) sites. Formal linkages were most effective where they had facilitated, and were supported by, informal networking. Informal networking was driven by the mutual benefits of information sharing and was optimised where sites were well aligned in terms of technology, geography and culture. Misalignments that created barriers to networking between organisations in a few cases were attributed to inappropriate choice of partners. Policy makers seeking to promote learning through centrally directed mechanisms need to create a framework that enables networking and informal knowledge transfer, allowing local organisations to develop bottom-up collaboration and exchanges, where they are productive, in an organic manner.

## Introduction

Digitisation of healthcare systems is now central to many national policies to address the challenges associated with ageing populations, rising demands and the pressure to deliver high quality care in tightening economic climates [1]. National digitisation attempts in England have included the 2005 National Programme for Information Technology, characterised by central procurement of technological solutions, and, after some years of uncoordinated adoption of digital technologies by individual hospital sites, the 2017–2021 Global Digital Exemplar (GDE) Programme [2, 3]. The GDE Programme focuses on promoting inter-organisational learning to improve the pace and reduce the costs of digital transformation across the English National Health Service [4].

Given the limited national budget and uneven digital maturity of provider organisations, the GDE strategy involved supporting relatively digitally-mature acute, mental health and ambulance provider organisations (hereafter GDEs) to achieve internationally recognised levels of digital excellence and thereby serve as exemplars for the wider NHS. GDEs were paired with somewhat less digitally-mature Fast Follower sites (hereafter FFs) to promote knowledge transfer [5]. This would be facilitated by the production of 'Blueprints'–formal documents capturing learning on the implementation of digital systems in specific areas. Shared learning in

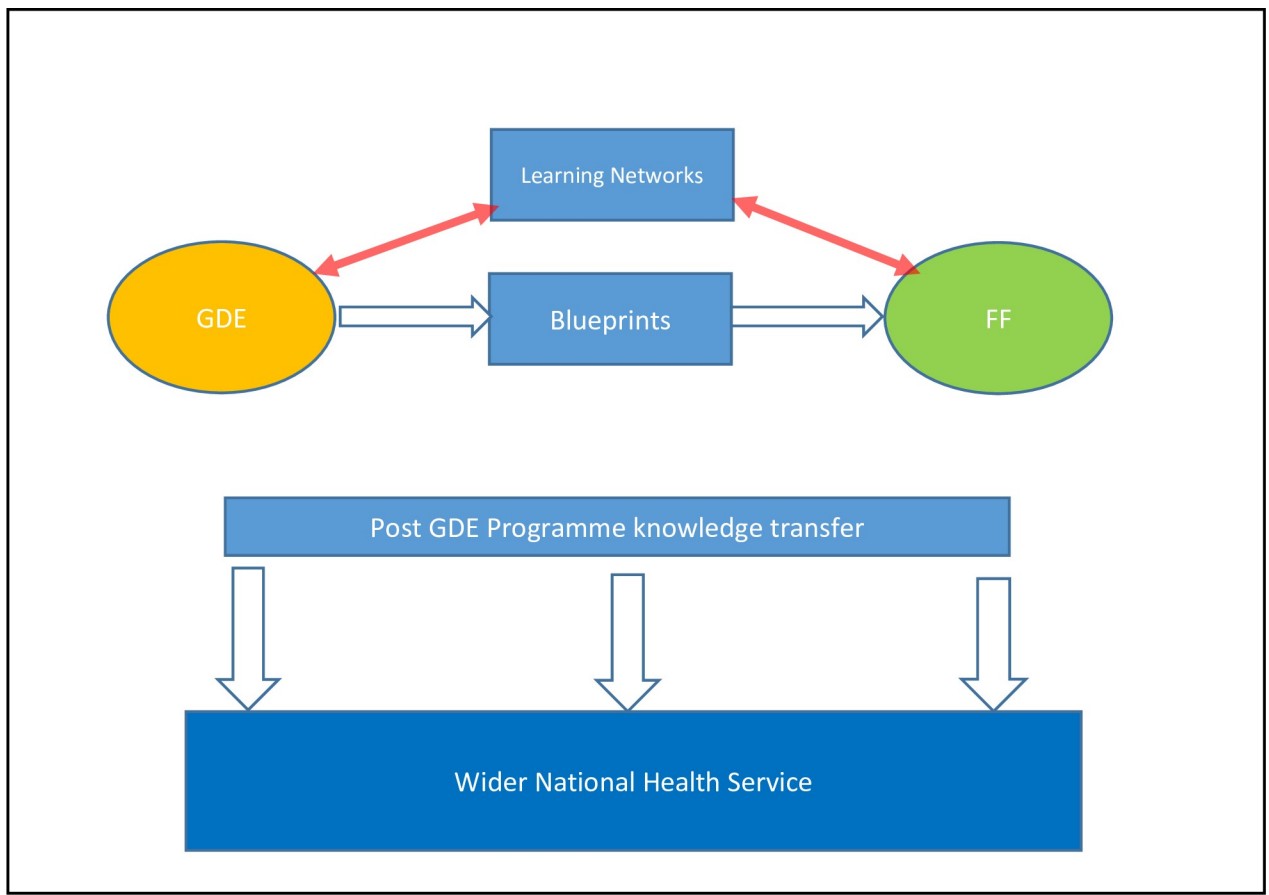

**Fig 1. Intended mechanisms of knowledge transfer between GDEs and FFs and to the wider NHS.**

the Programme was also facilitated by Learning Networks launched in the first half of the Programme, bringing together GDE and FF team members with an interest in specific topics. Though most ceased when funded was ended, the most successful ones, mental health and Hospital Electronic Prescribing Medicines Administration (HEPMA) became self-sustaining (see Fig 1).

There is currently a limited literature around inter-organisational knowledge transfer, particularly within health systems [6–11]. Existing scholarship has mainly focused on the role of system vendors and their users, and therefore may not readily be transferable to health systems [12, 13]. The GDE Programme therefore presents an important opportunity to examine these processes and this has potential international lessons for digitisation programmes in health systems.

This paper examines how the formal pairing of organisations, establishing partnerships between exemplar (GDE) and follower (FF) sites, has promoted knowledge transfer and supported digital transformation between paired organisations within the national GDE Programme. In doing so, we build on related work in progress that examines the role of other formal and informal inter-organisational knowledge sharing established as part of the GDE Programme [14]. We draw on perspectives surrounding sociotechnical systems, which assume that technological and social factors are interrelated and shape each other in complex ways in order to explore this issue [15].

## Methods

The study reported here is part of an independent longitudinal formative evaluation of the GDE Programme [16]. Detailed methods have been described in a published protocol [17]. This paper thus focuses on describing the particular aspects of the work that examined GDE/FF relationships.

### Ethical approval and permissions

This work was classified as a service evaluation and the study therefore did not require NHS Research Ethics Committee approval. Following standard practice for studies that fall outside of the remit of NHS National Research Ethics Service, we obtained ethical approval from The University of Edinburgh's School of Social and Political Science Research Ethics Committee (27.11.2017). Participants gave written informed consent.

### Sampling of provider organisations and respondents

We conducted a combination of interviews, observations and documentary analyses in 12 in-depth case study sites, 8 GDEs (6 acute and 2 mental health providers) and 4 FFs (3 acute and 1 mental health provider) and collected further data in 24 additional case study sites (15 GDEs, 9 FFs). Twelve FFs in the GDE Programme were not included in the study, comprising nine sites that joined the programme later and three that merged with their GDE in the course of the Programme. All GDEs participating in the Programme had relatively high levels of digital maturity, and in most cases embarked on major upgrades in core information infrastructures such as Electronic Health Record (EHR) implementation as part of the Programme.

Researchers purposefully sampled members of local GDE management teams who had knowledge and insights into the digital strategy and digital systems used at that site, and who represented various different professions and backgrounds, including Chief Information Officers (CIOs), Chief Clinical Information Officers (CCIOs), Programme Managers, and Project Leads. Our initial point of contact was the local GDE Programme Manager. In the in-depth case studies we also used snowball sampling to increase the size and diversity of our sample of respondents by requesting interviewees to recommend further participants, asking explicitly for those with differing involvements and varied viewpoints on the GDE Programme.

### Data collection

The research team comprised five researchers with backgrounds in science, technology and innovation studies and policy research, and four researchers with operational experience from NHS consultancy.

Data collection consisted of semi-structured interviews (see Table 1 for topic guide) and non-participant observations exploring perceptions of relationships and knowledge flows, and collection of documents to explore strategic plans. We also conducted semi-structured interviews with national policy makers and analysed national strategy documents to explore national mechanisms, and attended a range of conferences and workshops to gain insights into national plans and dynamics.

Interviews were digitally recorded and transcribed verbatim by a professional transcribing service.

### Data analysis

We uploaded transcripts, documents and observation notes into qualitative analysis software NVivo 11 and thematically analysed them, applying both deductive and inductive methods. Four researchers formulated a coding framework based on the existing literature, the questions

**Table 1. High-level interview topic guide focusing on exploring the Global Digital Exemplar/Fast follower relationship.**

| |
|---|
| • Background and role of interviewee(s) |
| • Implementation strategy and benefits realisation strategy |
| • New digital functions being introduced as part of GDE programme and other current/recent changes |
| • Overall thoughts on GDE Programme |
| vBenefits realisation and reporting |
| • Blueprints |
| • Relationship with vendors |
| • Knowledge management, networking and learning (formal and informal) |
| • What do you think was most important factor that contributed towards learning and knowledge exchange to help achieve Global Digital Exemplar (GDE) programme aims? |
| • Relationship between the Fast Follower (FF) and GDE site? |
| • Other relationships/sources of information, e.g. international partners; other sites. Can you describe any collaboration and work that you are doing with other GDE or fast follower sites? |
| • What can be done on a national level to promote the most effective networks? |
| • How can central expertise support digitisation beyond GDE? |
| • Lessons learnt and way forward |

asked in the interview topic guide and their knowledge of the GDE Programme. The coding framework evolved in line with emerging findings. Data were initially analysed within cases focusing on GDE/FF arrangements and knowledge transfer and then aggregated across cases, looking for confirming and disconfirming evidence. In doing so, we identified under what circumstances knowledge transfer worked best and what barriers existed. This process involved multiple rounds of detailed review and commentary by the whole evaluation team.

## Results

Our data consisted of 508 interviews, 163 observations, and analysis of 325 documents in 36 provider organisations. A detailed breakdown of these sources has been published in a previous paper [19].

Table 2 in S1 Appendix summarises the GDE and FF pairings, highlighting their system vendors, their local institutional context (whether they are part of the same Integrated Care System or Sustainability and Transformation Partnerships), prior relationships, and noting institutional mergers and other significant local institutional changes such as merging procurement departments. Of the 23 GDE and FF pairings, 17 used the same core technological systems. Four provider organisations adopted a Best-of-Breed approach (where different products from different vendors are connected via application processing interfaces). Two pairings had different systems. Many (14) GDE/FF pairings were within the same regional coordinating structures. (Sustainability and Transformation Partnerships [STPs] which include all health and social care providers within a geographical area, and or Integrated Care Systems [ICSs] which are a similar group of providers over a larger geographical area) [18]. Six organisations were involved in mergers in the course of the GDE Programme (five with their GDE/FF partner and one with another local hospital).

Fig 2 provides an overview of factors promoting informal networking between GDE and FF sites.

### Enhanced learning and accelerated adoption of technologies

We found that most interviewees believed the formally established GDE/FF relationship had enhanced knowledge exchange and accelerated adoption of technologies. Participants stated

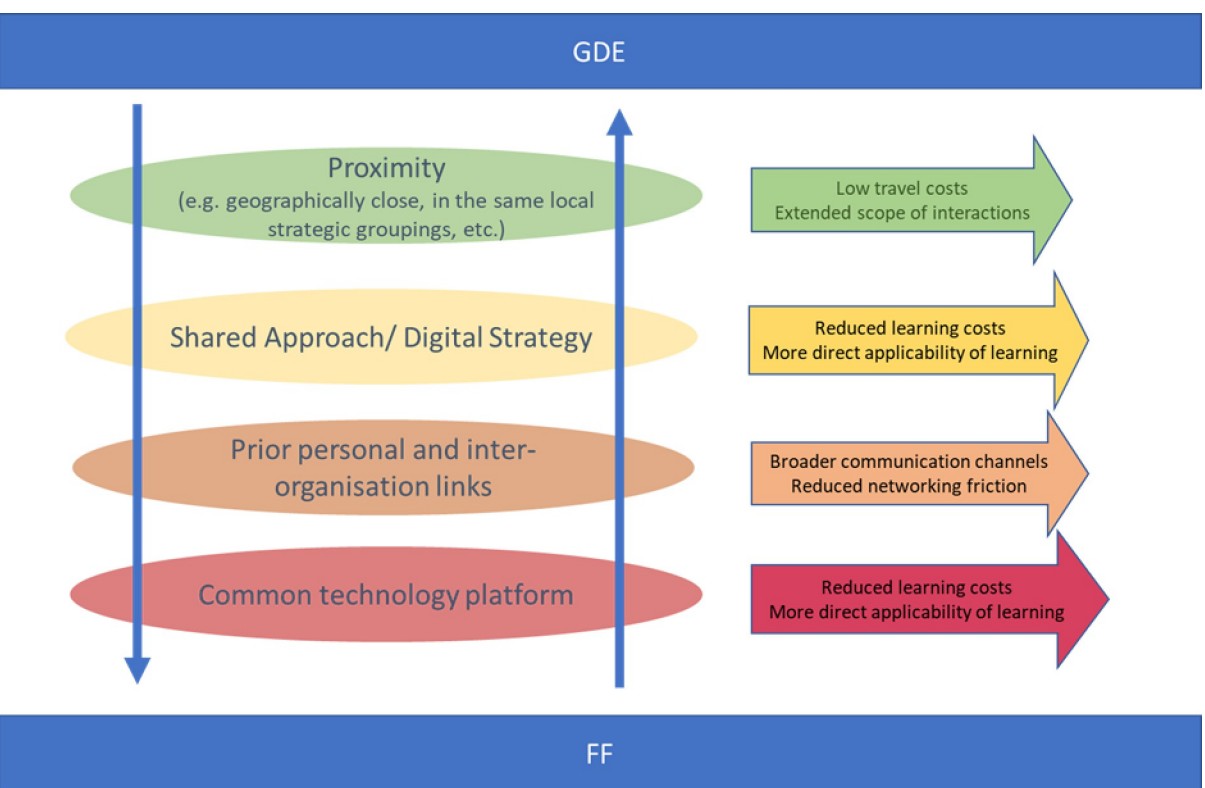

**Fig 2. Factors promoting informal networking between Global Digital Exemplar and Fast Follower sites.**

that this knowledge transfer was not just about technical matters–it included, for example information governance, training, change strategies, care pathways and advice on clinical engagement.

> *Certainly, in our experiences with [Fast Follower] is they would say they have learned a lot in terms of the way we use clinical support, the way we do testing. . .so they learned a lot from our groups (Site D case study, GDE Programme staff)*

Respondents highlighted the time and cost saving resulting from the GDE/FF relationship. Rather than starting from scratch, sites felt able to take on board solutions developed by their partners in the knowledge that these solutions had proved safe and effective in similar organisation.

> *And we didn't spend weeks and weeks reviewing it, we spent, you know, a two hour session understanding, with the right people in the room, what (GDE) did. . .And it's taken them five years to develop it and we did it in, you know, in one year. (Site L case study, FF, clinical digital leader)*

Though the GDE Programme's terminology suggests a one-way flow of information from exemplar (GDE) to follower (FF), most respondents pointed to two-way knowledge transfer, with GDEs also learning from their FFs.

> *So I sometimes, jokingly, call [FF partner] our fast forwarder. Because they, to me, are new eyes on things that we have, and they see things differently, and have suggested places we could improve our solution. (Site AG20, broader study, GDE, non-clinical digital leader)*

Some FFs were not happy with the label Fast Follower where they did not see themselves as lagging behind in competence and capability and it therefore did not reflect the actual relationship. One FF noted that they were ahead of their GDE in implementing the same EPR system.

*We had already been live with our EPR [electronic patient record] system for, it must have been, two years. . .in fact, I don't even think they'd signed the contract at that point, so they hadn't implemented their EPR but yet we were following them. So it was a strange marriage, we'll say. (Site J case study, FF, non-clinical digital leader)*

*Because we're moving forward, aren't we. . .I'm not sure I want to follow. And I think we want to be alongside with them. (Site F, case study, GDE, clinical digital leader of FF)*

In another case, participants stated that knowledge was not effectively shared between a GDE and FF adopting the same system as the FF was implementing a newer version of the package and did not feel they had much to learn from their GDE.

It was also highlighted that in some cases, the GDE/FF relationship resulted in cycles of improvement where FFs tested a newer version of the system and this in turn had the potential for GDEs to save valuable time when implementing the same upgrade.

*[Our FF] went live with a system that was much more developed, and it was two years more up to date. . .. So we were able to share now and look into the content that they had, and we can copy that back in. So it increases and accelerates our ability to keep up to date. (Site AG 8, broader study, GDE, clinical digital leader)*

## Uneven impacts of formal GDE/FF arrangements

However, we also observed a few sites where the formal programme pairing arrangements were perceived as less effective. These sites expressed concern about how their GDE/FF combinations had been chosen. These pairings had been set up under time pressures resulting from the short timeframes in which the GDE Programme had been developed and launched. Participants reported that sites generally sought to establish partnerships with organisations they were already collaborating with. However, this was sometimes perceived to conflict with the programme strategy, which, for example, encouraged partnerships between sites using the same core platform. In addition, acute hospital providers were only allowed to pair with other acute hospitals, and mental health services with other mental health services. Thus, the CIO of an acute GDE, partnered with an FF using the same platform but two hours' drive away, would have preferred to have a local mental health provider, which they later merged with, as their FF.

*So we wanted to look at our community mental health trust [provider organisation] as a Fast Follower, rather than another acute trust. . . So that was our preferred route as a Fast Follower because we could see the benefits of integration and how you could tell a story of an integrated healthcare system. But unfortunately that didn't fit the model of, you couldn't have a community mental health fast-follower, to an acute trust because it didn't fit the GDE model. (Site D case study, GDE, non-clinical digital leader)*

Where the GDE/FF pairing did not emerge from existing links, interviewees highlighted that there was a need to build relationships with consequent greater uncertainty about outcomes.

*And the truth is, it's not worked anything like as well as [additional FF], has it?. . .I think that was partly because we just didn't know the people there at the outset. . .it just meant that things didn't get done that might have got done otherwise. (Site AG13, broader study, GDE, clinical digital leader)*

Although the design of the GDE Programme conceived the GDE/FF relationship as revolving around the production of and adoption of Blueprints, there was little evidence in participants' accounts that these were a significant channel for knowledge transfer between the GDE and FF. One reason highlighted was that GDEs were so busy implementing new systems they did not initially have time to write Blueprints, which were produced at a later stage. Knowledge was instead transferred between the GDE and FF through direct contacts: site visits, phone calls and videoconferences and other electronic exchanges, and/or attending each other's committees. Participants perceived these to be a more effective vehicle for sharing and support than a formal Blueprint document [19].

*I haven't seen a Blueprint from* [our GDE] *for example,* [they] *don't have a Blueprint for [specific application] yet, as far as I'm aware, I haven't seen one, although we are creating one ourselves. (Site M case study, FF, GDE programme staff)*

### Enablers and barriers to organic knowledge transfer between GDEs and FFs

Knowledge transfer, and in particular the explosion of informal networking, was according to participants driven most immediately by the benefits participants derived from exchanging knowledge and experience with their peers. By examining perceived variation in the experience and effectiveness of knowledge exchange between sites, we can identify various enabling and inhibiting factors at play. The uneven contours of informal networking described by interviewees reveal the factors that enhanced the benefits and reduced the learning and coordination costs of knowledge sharing.

**Shared technological platform.** Where an FF had the same core technology platform as its GDE (e.g. EHRs and Hospital Electronic Prescribing and Medicines Administration [HEPMA] Systems), learning was viewed to be more readily applied and offer greater benefits as sites could readily adopt elements of their solutions (including system configurations and workflows which had often been arduous to produce) without much need to amend them.

*So we are Fast Followers to* [named GDE]. *Specifically, truly the real fast following with [this site] is about ePrescribing. So the whole HEPMA project. We have worked extremely closely with them. We have more or less cut and pasted all their workflows, all their pharmacy workflows, all their drug administration workflows. . . we've actually paid for time of their lead project pharmacist. They have attended all our design workshops in the early days. . .without that involvement, the project would have taken longer. . .I think the result is safer and more robust than it would have been if we had done it without their help. (Site L, case study, FF, senior manager)*

**Geography.** According to participants, many of the GDEs had selected FFs that were in close proximity. This was stated to be useful in terms of reducing the time and money costs of travel. It thus also facilitated more intense forms of collaboration according to participants. They reported that one GDE/FF partnership decided to create a joint procurement team as a result of their successful collaboration. In another provider organisation, an interviewee

reported that the proximity of the GDE site meant a clinician could come over and test their system.

> *And then luckily for us we have one of the clinicians working on our site on Tuesdays and Thursdays. . .. So I've given her access to our system, our test systems, for her to just go in and test and then see where we need to improve upon, because they've used it for quite some time. . .. So it's like lessons learnt. So she's been really, really helpful.* (Site M case study, FF, GDE programme staff)

Participants highlighted that proximity was also associated with other enabling factors related to knowledge transfer, including inter-personal (see below) and institutional linkages. They reported that nearby sites were often within the same STP/ICS–the emerging regional coordination structures, which have become increasingly salient in the course of the GDE Programme. These institutional linkages were seen to help in developing a common digital strategy and broader outlook.

That said, we also found evidence of successful GDE/FF partnerships at greater distance. Geography was not seen as a barrier when the benefits of learning and sharing were perceived to be substantial, with networking often facilitated by other enablers such as prior collaborations, interpersonal relationships, similarity of platform and a shared philosophy of sharing for the benefit of the NHS.

> Non- clinical digital leader: *I don't know that there are advantages. I mean it would be interesting to work with a trust [provider] that we haven't worked with up till now. Obviously, you haven't got the STP, You haven't got the local structures to make that make sense. But in actual fact we work remotely most of the time from [site]. So the physical nearness is perhaps less important than we would've thought two years ago.*
>
> Senior manager: *Rather than the enthusiasm of somebody that actually wants to work with you, which I think is very important. (Site L case study, FF, senior manager and non-clinical digital leader)*

**Peer-to-peer prior relationships.** Proximity was also perceived to be related to the greater likelihood of prior linkages between the individuals and groups in the organisations involved. Participants reported that some interpersonal relationships of key staff resulted from previous experience of working together or from staff movements between sites. In the case of Site M, the project manager for implementation of the Clinical Data Repository (CDR) had previously worked on the same project for the GDE. At Site F, the CIO already knew staff at the FF site some distance away. Some relationships were reported to be based on pre-GDE collaborations. One respondent observed that these kinds of links could encourage greater openness to external ideas.

> *I think with the Blueprints, no matter how good they are you've still got a locked door of people who will want to come up with it themselves and you have to change that mind-set there. And I think you do that by getting people moving around. (Site H case study, GDE, senior manager)*

## Discussion ### Summary of main findings

Participants reported that most GDE/FF pairings resulted in enhanced inter-organisational knowledge transfer and accelerated technology adoption in participating organisations. They were seen to be most effective where they were buttressed by a growth in informal networking

that was driven by the mutual benefits of knowledge sharing. Perceived variations between sites in the intensity of informal networking highlighted incentives and barriers at play. Thus, participants reported that the benefits of knowledge sharing were enhanced where there were common technological platforms and comparable context. Physical proximity and prior linkages were stated to reduce, respectively, the travel and coordination costs of networking. In contrast to the Programme's terminology that projected a one-way flow of knowledge from Exemplar to Fast Follower, knowledge transfer was seen to be bi-directional, characterised by reciprocal and ongoing exchanges. Sites felt a partnership model would have been more effective.

## Strengths and limitations

We have collected a wealth of qualitative data from different sources over an extended timeframe to get contemporary insights into the formal knowledge sharing processes put in place nationally in the English National Health Service by pairing organisations to share digital transformation knowledge. Although the longitudinal qualitative nature of data collection has allowed us to gain insights into unfolding relationships over time, it has not been sufficient to allow us to link identified processes to implementation outcomes. This shortcoming reflects a general issue with complex transformational programmes, where outcomes emerge gradually and are often difficult to attribute [20]. This work is based on insights derived from evaluating the first 24 months of the GDE Programme. A further round of fieldwork has been paused due to the unfolding COVID-19 pandemic, limiting the longitudinal base of data collection and thus also the scope to derive insights into the further evolution of these processes and their sustainability. The Evaluation sought to analyse knowledge flows amongst a web of actors from the vantage point of case studies of provider organisations. This gave us a strong opportunity to explore knowledge transfer between paired (GDE/FF) organisations and how this was influenced by informal networks and other factors (as summarised in this paper). This study, and a related investigation into other mechanisms of knowledge transfer through the production and circulation of Blueprints, are part of an attempt to examine how the GDE Programme has sought to establish a learning ecosystem. The latter are the subject of other publications [14, 19]. A further limitation related to the sampling of participants and case study sites. Many of the sampled participants may, due to their role as local programme managers, have been biased towards successful accounts of the Programme. We collected more data in GDEs than in FFs, which may have biased overall perspectives. FFs joined the programme later and were thus under- represented in the detailed longitudinal in-depth studies. We did not collect data on the number of electronic health records and other digital systems implemented by providers outwith the GDE Programme in this period. We can therefore not make any quantitative claims on the effectiveness of knowledge transfer. In almost all of the GDE and FF pairings the knowledge transfer was bi-directional. We cannot add exceptions relating to specific providers as this would highlight the cases where this did not happen and this would compromise our confidentiality agreement with provider organisations.

## Placing the work within the wider empirical literature

This work contributes to the currently sparse literature around inter-organisational knowledge sharing in digital transformation in healthcare settings [6, 9]. It illustrates that partnering organisations with a shared technology focus can help to facilitate knowledge transfer as lessons and even parts of solutions developed in one site can be readily and beneficially reused. Geographical proximity reduces the travel costs of face-to-face interactions and is often associated with prior inter-personal and inter-organisational linkages and cultural alignments, which have also been shown to promote informal networking, consistent with findings in other contexts [21–24]. Informal networks–which often operate independently from formal structures

and cannot be predictably established by top-down plans–have an important role in promoting knowledge transfer [25–27]. Knowledge sharing through direct interactions and informal networking has a particular value in sharing tacit knowledge (knowledge that is difficult to codify) and translating experiences and lessons from one setting to another through "learning-by-interacting" [28, 29].

Inter-organisational knowledge sharing is characterised by a variety of informal as well as formal networks that change over time [30]. Their operation is shaped (facilitated or inhibited) by the social and organisational contexts [31]. Existing relationships and the alignment of institutional contexts can reduce barriers and maximise incentives–in particular the mutual benefits of shared learning–establishing a virtuous cycle through which knowledge sharing is effective and sustainable. This can be promoted by formal means, but there is unlikely to be a universal "recipe for success" as learning and knowledge sharing is an unpredictable process. Knowledge transfer between organisations does not operate in isolation and needs to be seen in the context of the wider ecosystem [32]. These findings from studies in the sociotechnical systems tradition are illustrated by GDEs and FFs in our sample that were sharing knowledge with other provider organisations outwith their formal pairings.

## Implications for practice and policy

Our work shows that formal inter-organisational partnerships can provide an effective mechanism for knowledge sharing and collaboration, particularly where buttressed by informal knowledge networking. This is further reinforced by the overall success of the Programme in stimulating digitally enabled transformation and the sharing of knowledge between participating provider organisations. GDE/FF partnerships were one knowledge sharing mechanism that helped, together with others e.g. Blueprinting and Learning Networks, to promote an ethos of shared learning to promote Programme aims and learning in the wider NHS.

Where networking allowed lessons and solutions from other sites to be reliably re-used, the saving in time and effort could reduce the costs and increase the speed of change. Informal knowledge networking, driven primarily by the mutual benefits of knowledge sharing, was encouraged by common technological systems (offering more immediate applicability of knowledge, experience and solutions), by geographical proximity (though effective knowledge transfer also arose in some circumstances between geographically dispersed organizations [33] and by emerging regional coordination structures, which were also associated with interpersonal links and shared culture that provided additional communication channels and facilitated collaborative culture.

The strategic partnerships observed in our work were formed through various mechanisms and channels of communication, both formal and informal. Knowledge sharing is therefore difficult to plan and may have many unanticipated benefits and/or difficulties. We found that, while flows of knowledge can be promoted and channelled to some degree through formal means, strategic decision makers need to be mindful of the importance of bottom-up knowledge exchange, driven by the benefits of sharing, which often follows different paths than planned knowledge transfer. Support should seek to align formal support with organic, bottom-up networking to achieve the mutual strengthening of both. This may for example be achieved through giving sites a degree of choice of partners they consider appropriate to their current organisational strategies at any point in time.

## Conclusions

The partnerships established under the GDE/FF programme helped to promote collaboration and knowledge transfer between participating sites to achieve the shared goal of improved

patient care and improved clinician experience and contributed to the overall success of the Programme. This study has also illustrated the unpredictability of knowledge flows and highlighted the importance of informal knowledge exchanges driven by the mutual benefits of knowledge sharing. There are important lessons for healthcare digitisation programmes seeking to promote knowledge sharing to accelerate technology implementation. Sharing core architectures and geographical proximity may facilitate informal networking that synergises with formal mechanisms and encourages the establishment of a broader inter-organisational digital health learning ecosystem.

## Supporting information

**S1 File.**
(PDF)

**S1 Appendix. Table 2 GDE/FF pairings, systems, local institutional contexts and mergers.**
(DOCX)

## Author Contributions

**Conceptualization:** Kathrin Cresswell, Aziz Sheikh, Bryony Dean Franklin, Robin Williams.

**Formal analysis:** Susan Hinder, Hung The Nguyen.

**Funding acquisition:** Kathrin Cresswell, Aziz Sheikh, Bryony Dean Franklin, Marta Krasuska, Robin Williams.

**Investigation:** Susan Hinder, Marta Krasuska, Hung The Nguyen, Wendy Lane, Kathy Mason, Sally Eason.

**Methodology:** Kathrin Cresswell, Aziz Sheikh, Bryony Dean Franklin, Wendy Lane, Hajar Mozaffar, Kathy Mason, Sally Eason, Henry W. W. Potts, Robin Williams.

**Supervision:** Robin Williams.

**Writing – original draft:** Susan Hinder.

**Writing – review & editing:** Kathrin Cresswell, Aziz Sheikh, Bryony Dean Franklin, Marta Krasuska, Hung The Nguyen, Wendy Lane, Hajar Mozaffar, Kathy Mason, Sally Eason, Henry W. W. Potts, Robin Williams.

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
