## [Decision Letter · Decision Letter 0]

25 Feb 2021

PONE-D-20-35004

Promoting inter-organisational knowledge sharing: a qualitative evaluation of England’s Global Digital Exemplar and Fast Follower Programme

PLOS ONE

Dear Dr. Hinder,

Thank you for submitting your manuscript to PLOS ONE. After careful consideration, we feel that it has merit but does not fully meet PLOS ONE’s publication criteria as it currently stands. Therefore, I invite you to submit a revised version of the manuscript that addresses the points raised during the review process.

We look forward to receiving your revised manuscript.

Kind regards,

Shahrokh Nikou

Academic Editor

PLOS ONE

Additional Editor Comments:

Thank you for your submission to the PLOS ONE. We greatly appreciate the time and energy you devoted to preparing your manuscript.

We were fortunate to have three highly qualified reviewers for your manuscript that provided insightful and timely reviews across the theory and method employed in this manuscript as well as the contributions to the. Overall, the reviewers were quite about some elements of the manuscript.

While, reviewers believe the manuscript discusses an important topic, however, the reviewers also highlighted a number of serious and important concerns that prevent the manuscript from reaching its potential. In sum, the reviewers were quite mixed in their evaluations of the manuscript. Please, see comments below.

Journal Requirements:

"All authors are investigators on the evaluation of the Global Digital Exemplars Programme (http://www.ed.ac.uk/usher.

AS was a member of the Working Group that produced " Making IT Work" and was an assessor in selecting GDE sites. BDF supervises a PhD partly funded by Cerner, unrelated to this paper"

4. We noted in your submission details that a portion of your manuscript may have been presented or published elsewhere.

"Only the table describing the number of Interviews undertaken for the evaluation study"

Reviewers' comments:

Reviewer's Responses to Questions

**Comments to the Author**

1. Is the manuscript technically sound, and do the data support the conclusions?

Reviewer #1: Yes

Reviewer #2: Yes

Reviewer #3: Partly

2. Has the statistical analysis been performed appropriately and rigorously? 

Reviewer #1: N/A

Reviewer #2: N/A

Reviewer #3: N/A

3. Have the authors made all data underlying the findings in their manuscript fully available?

Reviewer #1: Yes

Reviewer #2: Yes

Reviewer #3: Yes

4. Is the manuscript presented in an intelligible fashion and written in standard English?

Reviewer #1: Yes

Reviewer #2: Yes

Reviewer #3: Yes

5. Review Comments to the Author

Reviewer #1: This is a well crafted and well written paper assessing lessons to be drawn from efforts to promote the digitalisation of hospital services in England. The methodology is clear. The analysis has a good structure and makes excellent use of quotations. Based on multiple interviews and observations, the case for paying heed to informal knowledge networking and the potential benefits of two-way flows of information is well made and based on an interesting exposition of the evidence.

A few minor revisions might be helpful.

1. The abstract indicates that the research draws on socio-technical systems theory. The introduction might explain what this is and how it orients the research in a particular way.

2. Figure 1 comes at the end of the discussion. Something should be said about the figure in the text, i.e. it is a summary ….

3. Line 360 where references are made to the literature – should the text read something like ‘networking, consistent with findings in other contexts’ since the literature is not specifically about health systems. The subtitle indicates it is a discussion of the wider literature but it would help to clarify here.

4. Line 372 should the sentence starting ‘This is illustrated …’ read something like. These findings from studies in the socio-technical systems tradition are illustrated by GDEs and FFS in our sample that were sharing knowledge …

5. Line 385 ‘These strategies partnerships …’ Here is this in reference to your empirical study partnerships or is this statement made in relation to similar partnerships. Unclear whether the discussion in this section is meant to refer to this study primarily or to generalisations from it to similar situations.

Small corrections

Line 279. Should it read ‘the GDE site meant a clinician [came or could come] over and could test their system’

Line 308 greater likelihood [of] priori linkages

Reference 29. Citation seems incomplete.

Reviewer #2: Thank you for this interesting paper.

It is interesting to have some insights into the GDE program.

The study is well -executed.

I am interested in several points:

1.What is the digital maturity of the participants? The complexity and sophistication of the knowledge transfer regarding implementing an EMR is quite different to sharing knowledge for implementing a basic LIS.

It was difficult to truly understand the program from the text. Would it be possible to represent the program diagrammatically? Map out the realtionships of the GDEs and FFSs and perhaps indicate the size and digital maturity of each facility in the diagram and the nature of the relationships. How many were positive, negative, neutral? Which ones had unidrectional or bidirectional information flows? This would allow a more detailed and granular understanding of the program, particularly for international readers.

2.Did digital maturity across the NHS as a whole increase during the GDE program? I take the author's point that it is impossible to attribute any granular outcomes to the program, however I think it is important to understand did the NHS as a whole complex adaptive system, continue to undertake digital transformation ? How many EMRs were implemented during the GDE program? How many other applications were implemented? It would be critical to know, for example , if the number of EMRS deployed actually decreased or stalled during the program. Conversely, it would strengthen the conclusion and discussion of the paper to know how many systems were implemented in the FFS during the program. There is no data to suggest how many applications were actually adopted by the FFs during the program, which as I understand it, was the primary intent of the program. IF interorganisational knowledge transfer was effective, we would at least expect ongoing implementation of new systems in the FFs. and this should be reported.

Minor point: be consistent on the venison of NVivo used, it changes throughout text

Reviewer #3: Thank you for the opportunity to review this qualitative evaluation of the Global Exemplar and Fast Follower Programme in England. This programme is of interest to many jurisdictions across the world.

My main comments relate to some of the overly quantitative language used to present qualitative findings and the methods utilised for sample selection. Some clarification/revision here may help acceptance and translation of this work.

Specifically:

Abstract

-The results state that partnerships enhanced learning and accelerated adoption. However the study does not attempt to answer this question. It assesses users' opinions of the pairings, and there is no comparison to non-paired arrangements. The conclusions and language in the actual paper is more moderate and I think the abstract should reflect this.

Methods

The study deliberately sampled members of local GDE teams via the GDE program manager. Presumably many (if not most) of these individuals' jobs are funded by- and dependent on the success of this program. How was this potential conflict of interest and bias in sampling managed?

Data collection(Table 2). The table outlines the type of data collection methods. However we are not shown the proportion of data collected (interviews, meetings, etc)from GDE v FF. Given that there were twice as many in depth case study sites that were GDEs(p5 Line 96), is this sample disproportionately from a GDE perspective? Again a perspective that might be more likely to be positive.

6. PLOS authors have the option to publish the peer review history of their article (what does this mean?). If published, this will include your full peer review and any attached files.

Reviewer #1: No

Reviewer #2: No

Reviewer #3: **Yes: **

---

## [Author Response · Author response to Decision Letter 0]

10 May 2021

Responses to all reviewer and editor comments are in the Cover Letter

---

## [Decision Letter · Decision Letter 1]

8 Jun 2021

PONE-D-20-35004R1

Promoting inter-organisational knowledge sharing: a qualitative evaluation of England’s Global Digital Exemplar and Fast Follower Programme

PLOS ONE

Dear Dr. Hinder,

Thank you for submitting your manuscript to PLOS ONE. After careful consideration, we feel that it has merit but does not fully meet PLOS ONE’s publication criteria as it currently stands. Therefore, we invite you to submit a revised version of the manuscript that addresses the points raised during the review process.

We look forward to receiving your revised manuscript.

Kind regards,

Shahrokh Nikou

Academic Editor

PLOS ONE

Additional Editor Comments (if provided):

While one of the reviewers is happy with the revision, the second reviewer still some issues.

Therefore, I invite the authors to address the comment stated below.

Reviewers' comments:

Reviewer's Responses to Questions

**Comments to the Author**

1. If the authors have adequately addressed your comments raised in a previous round of review and you feel that this manuscript is now acceptable for publication, you may indicate that here to bypass the “Comments to the Author” section, enter your conflict of interest statement in the “Confidential to Editor” section, and submit your "Accept" recommendation.

Reviewer #2: All comments have been addressed

Reviewer #3: (No Response)

2. Is the manuscript technically sound, and do the data support the conclusions?

Reviewer #2: (No Response)

Reviewer #3: Partly

3. Has the statistical analysis been performed appropriately and rigorously? 

Reviewer #2: (No Response)

Reviewer #3: N/A

4. Have the authors made all data underlying the findings in their manuscript fully available?

Reviewer #2: (No Response)

Reviewer #3: No

5. Is the manuscript presented in an intelligible fashion and written in standard English?

Reviewer #2: (No Response)

Reviewer #3: Yes

6. Review Comments to the Author

Reviewer #2: (No Response)

Reviewer #3: Thank you for the opportunity to review this revised manuscript and the associated response to reviewer comments document.

Although you have attempted to address most of the comments, I feel these revisions neds to go further to adequately address the comments.

With respect to the comment about presenting qualitative findings as quantitative conclusions(especially in the abstract), this has been addressed in the results summary in the abstract but the conclusion. This type of language still exists throughout the main manuscript, even though the limitations section has been expanded.

With respect to the snowballing and targeting negative perceptions in the methods, how was this done? How would a researcher at another centre reproduce the method to gain comparable results?

Figure 1 is a simplified schematic that in my opinion doesn't address the suggestion from reviewer 2 which I think would add a great deal to the paper. This aligns with reviewer 3 comment 3. It doesn't have to be a detailed analysis of the influence of size and maturity on the results, just a representation of these aspects and relationships of the study participants to better contextualise the findings.

7. PLOS authors have the option to publish the peer review history of their article (what does this mean?). If published, this will include your full peer review and any attached files.

Reviewer #2: No

Reviewer #3: No

---

## [Author Response · Author response to Decision Letter 1]

21 Jun 2021

Response to Reviewers

Reviewer 3 

Point 1: With respect to the comment about presenting qualitative findings as quantitative conclusions (especially in the abstract), this has been addressed in the results summary in the abstract but the conclusion. This type of language still exists throughout the main manuscript, even though the limitations section has been expanded.

Response: We have made clear in the Results and Discussion sections that the findings were based on user perceptions and not on quantitative measures of learning and adoption.

Point 2: With respect to the snowballing and targeting negative perceptions in the methods, how was this done? How would a researcher at another centre reproduce the method to gain comparable results? 

Response: We asked interviewees for recommendations of further participants to interview, asking explicitly for those with differing involvements and varied viewpoints on the GDE Programme. We have now made this clearer in the Methods section.

Point 3: Figure 1 is a simplified schematic that in my opinion doesn't address the suggestion from reviewer 2 which I think would add a great deal to the paper. This aligns with reviewer 3 comment 3. It doesn't have to be a detailed analysis of the influence of size and maturity on the results, just a representation of these aspects and relationships of the study participants to better contextualise the findings. 

Response: We have amended Figure 1 to represent the flow of knowledge from GDEs to FFs as intended in the GDE Programme and we now describe in more detail in the text how practice varied from this plan. We have, in addition, substantially reworked the detailed breakdown of GDE/FF pairings in Table 2 (Appendix). We have added a column noting any prior relationships between GDE and FF and indicate the size of the hospitals by including numbers of beds (where available). We have replaced specific details of local regional groupings with a Yes/No column on whether GDE and FF are in the same regional group. 

In almost all of the GDE and FF pairings the knowledge transfer was bi-directional. We cannot add this information relating to specific providers as this would compromise the agreed terms of access for the evaluation which included a firm commitment not to identify specific provider organisations or individuals. 

Finally, assessments of the digital maturity of providers across the NHS (based on self-assessments) are held centrally within the NHS and not publicly available. We did not collect data on the number of electronic health records and other digital systems implemented in the rest of the NHS during the time period of the GDE Programme. We are therefore unable to add this context to the paper and have noted this in the Discussion section as a limitation. 

We trust that these revisions are to your satisfaction and that we are now in a position to proceed with publication. Please do not however hesitate to contact me if you require any further details. 

With kind regards,

Susan Hinder (on behalf of the co-authors)

---

## [Decision Letter · Decision Letter 2]

13 Jul 2021

Promoting inter-organisational knowledge sharing: a qualitative evaluation of England’s Global Digital Exemplar and Fast Follower Programme

PONE-D-20-35004R2

Dear Dr. Hinder,

We’re pleased to inform you that your manuscript has been judged scientifically suitable for publication and will be formally accepted for publication once it meets all outstanding technical requirements.

Kind regards,

Shahrokh Nikou

Academic Editor

PLOS ONE

Additional Editor Comments (optional):

Reviewers' comments:

**Comments to the Author**

1. If the authors have adequately addressed your comments raised in a previous round of review and you feel that this manuscript is now acceptable for publication, you may indicate that here to bypass the “Comments to the Author” section, enter your conflict of interest statement in the “Confidential to Editor” section, and submit your "Accept" recommendation.

Reviewer #3: All comments have been addressed

2. Is the manuscript technically sound, and do the data support the conclusions?

Reviewer #3: (No Response)

3. Has the statistical analysis been performed appropriately and rigorously? 

Reviewer #3: (No Response)

4. Have the authors made all data underlying the findings in their manuscript fully available?

Reviewer #3: (No Response)

5. Is the manuscript presented in an intelligible fashion and written in standard English?

Reviewer #3: (No Response)

6. Review Comments to the Author

Reviewer #3: (No Response)

7. PLOS authors have the option to publish the peer review history of their article (what does this mean?). If published, this will include your full peer review and any attached files.

Reviewer #3: No

---

## [Editor Report · Acceptance letter]

22 Jul 2021

PONE-D-20-35004R2 

Promoting inter-organisational knowledge sharing: A qualitative evaluation of England’s Global Digital Exemplar and Fast Follower Programme 

Dear Dr. Williams:

I'm pleased to inform you that your manuscript has been deemed suitable for publication in PLOS ONE. Congratulations! Your manuscript is now with our production department. 

Kind regards, 

on behalf of

Dr. Shahrokh Nikou 

Academic Editor

PLOS ONE